# Epidemic-Prevention Measures and Health Management in a Nursing Home during the Coronavirus Disease 2019 Pandemic

**DOI:** 10.3390/healthcare11182535

**Published:** 2023-09-14

**Authors:** Shu-Ting Chuang, Mei-Hui Lin, Honda Hsu, Chia-Ming Chi, Yu-Ru Lee, Ya-Hui Yen

**Affiliations:** 1Department of Nursing, Taichung Tzu Chi Hospital, Buddhist Tzu Chi Medical Foundation, Taichung City 427213, Taiwan; shuting@tzuchi.com.tw; 2Department of Nursing, Tzu Chi University of Science and Technology, Hualien 970046, Taiwan; 3Taichung Tzu Chi Nursing Home, Buddhist Tzu Chi Medical Foundation, Taichung City 427213, Taiwan; chiamingchie19911221@gmail.com (C.-M.C.); x101023y@gmail.com (Y.-R.L.); 4Department of Accounting Information, Da-Yeh University, Changhua 515006, Taiwan; mhlin1011@gmail.com; 5Division of Plastic Surgery, Dalin Tzu Chi Hospital, Buddhist Tzu Chi Medical Foundation, Chiayi City 622007, Taiwan; hondahsu@yahoo.com.tw; 6School of Medicine, Tzu Chi University, Hualien 970374, Taiwan; 7Ph.D. Program in Healthcare Science, China Medical University, Taichung City 406040, Taiwan; 8Department of Nursing, National Chi Nan University, Puli Township 545301, Taiwan

**Keywords:** coronavirus disease 2019 pandemic mortality, reablement, older adult loneliness, nursing home

## Abstract

This study aimed to investigate the impact of epidemic prevention and isolation policies on residents’ health and well-being and assess the effectiveness of implementing intervention measures to maintain their quality of life. This mixed-methods research study involved a retrospective record review of residents’ daily life diaries and descriptive statistical analysis. Data were collected between March 2021 and June 2022, and epidemic-prevention measures were implemented using Taiwan’s Centers for Disease Control guidelines. Three interventions were developed to address residents’ health, social, and rehabilitation needs. Despite an overall infection rate of 10% at various times between 2021 and 2022, there were no reported outbreaks of nosocomial infections. The concept of reablement proved effective in helping residents maintain their independence and physical function, with a maintenance rate of 66.6%, thereby improving their quality of life. By implementing epidemic-prevention measures, we found that proper hand washing and the use of surgical masks were effective in controlling infections. Furthermore, the decline in physical function is a continuous and gradual process for older adults. Even under the restriction of social interaction, it is essential to incorporate rehabilitation plans into residents’ daily activities and encourage their active participation, as this promotes improved physical function and enhances their overall quality of life.

## 1. Introduction

The coronavirus disease 2019 (COVID-19) pandemic has had a far-reaching impact on the global healthcare industry and economy, leaving a profound mark worldwide. Its effects have been experienced universally, leading to extensive disruptions and significant challenges across daily life [1]. Since the onset of the pandemic, mainland China and Italy have been among the most severely affected areas. The elderly population accounts for more than 70% of the reported deaths in mainland China [2]. According to the Italian National Institute of Health, individuals over 70 years old have a COVID-19 fatality rate of more than 35%, which increases to as high as 41% for those over 80 years old [3]. In August 2020, 8991 deaths related to COVID-19 were reported in Canada, with 7028 (approximately 78.2%) being residents of long-term care institutions. Similarly, data from European and American countries reveal that 30% to 60% of COVID-19 deaths were residents in long-term care facilities [4,5]. During the COVID-19 pandemic period of 2020–2021, it was estimated that the global mortality rate was 2.75% [6]. Taiwan’s overall mortality rate was 0.17%, which was relatively low compared to many other countries. However, it is worth noting that 2.2% of deaths occurred in long-term care facilities, indicating that these facilities may be particularly vulnerable to the COVID-19 pandemic [7]. As of late April 2021, the estimated death rate for COVID-19 in nursing homes in the United States was 8.4% [8]. Nursing homes are recognized as having a high risk of COVID-19 infection due to their residents’ advanced age and often multiple chronic conditions. Data on morbidity and mortality rates from healthcare facilities indicate that older adults with chronic diseases and cardio-pulmonary conditions are particularly susceptible to new coronavirus infections [9,10,11,12]. This presents a challenge for infection prevention and control [13,14].

In January 2020, Taiwan’s Centers for Disease Control (CDC) classified COVID-19 as a Category Five notifiable infectious disease. Given the serious consequences of COVID-19 in long-term care institutions, management strategies for pandemic outbreaks in nursing homes can be particularly complex [15]. The prevention and outbreak control of COVID-19 infections were placed under the joint direction of Taiwan’s health and medical units to address this. Taiwan’s CDC released the “Guidelines on Infection Control Measures for Long-Term Care and Social Welfare Institutions in Response to Severe Special Infectious Pneumonia” [16]. The guidelines prioritize healthcare workers’ knowledge and training to tackle the ongoing epidemic efficiently. It is essential to equip healthcare workers with a comprehensive understanding of the current epidemic, enabling them to respond effectively. Moreover, the guidelines require the enforcement of social-distancing measures in long-term care institutions to minimize the risk of infection among residents. Additionally, it is essential to raise awareness among the public about practicing proper cough and respiratory hygiene, wearing masks, and frequently washing hands. Institutions must also enforce visitor restrictions to prevent the spread of the disease. Residents with fever must be isolated in one room, observed, and treated individually. Moreover, promoting vaccination among the populace is crucial to control the epidemic. Studies have shown that restricting family visits can effectively prevent the spread of disease. Nonetheless, social distancing and home isolation have significantly reduced physical activity, which can negatively impact older adults’ physical, psychological, and social health [17,18,19]. Some studies have shown that such measures can increase anxiety, fear, and loneliness among institutionalized residents and may exacerbate their feelings of abandonment [20,21].

During the pandemic, various preventive measures, such as restrictions on physical activities, limitations on social interactions, and isolation measures, may lead institutional residents to withdraw from life, resulting in the rapid deterioration of physical function and the onset of depression among older people. It is essential to prioritize optimizing opportunities for health participation and security to maintain the quality of life of aging adults. The World Health Organization (WHO) acknowledges the importance of prioritizing older adults’ physical, social, and mental well-being and advocates for their active engagement in daily activities. By providing opportunities for meaningful participation, older adults can experience improved overall health and enhanced well-being [22]. Additionally, it is crucial to maintain the physical activity function of residents to uphold their quality of life during the pandemic. Reablement training is a novel home-care approach that fosters a supportive and enabling environment to help individuals to engage in their daily activities and regain a sense of value and purpose [23]. This approach is customized to meet each person’s unique needs and seamlessly integrates into their daily routine. By creating a comfortable and supportive environment, residents are encouraged to participate in activities that they find meaningful, allowing them to improve their functional ability to perform daily tasks gradually. Reablement training helps individuals to achieve optimal functional status while participating in activities that enhance their overall quality of life [24]. Considering the above, during the pandemic, elderly care institutions must adhere to government preventive measures while implementing complementary measures to ensure the health and quality of life for older adults.

This study aimed to ensure residents’ quality of life and prevent COVID-19 outbreaks in a nursing home during an epidemic. To achieve this, we collaborated with the government to implement the “Guidelines on Infection Control Measures for Long-Term Care and Social Welfare Institutions in Response to Severe Special Infectious Pneumonia” provided by Taiwan’s CDC. To maintain the residents’ health and quality of life, we implemented three interventions addressing their health, social, and rehabilitation needs. These interventions included using health technology, video call equipment, and reablement as a substitute for daily group activities. Our research hypotheses were as follows: (1) the guidelines provided by Taiwan’s CDC are effective, (2) timely implementation of healthcare support measures is effective, and (3) replacing group activities with reablement can effectively maintain residents’ functional abilities. The study aimed to investigate the impact of epidemic prevention and isolation policies on residents’ health and well-being while evaluating the effectiveness of intervention measures in promoting their engagement in daily activities and preserving their quality of life.

## 2. Materials and Methods

### 2.1. Study Design

We used a mixed-methods research design that included both quantitative and qualitative analyses. A retrospective review of the residents’ daily life diaries and descriptive statistical analyses was conducted to collect demographic information. Data were collected between March 2021 and June 2022 to examine the incidence of COVID-19 and mortality among the residents, which was then compared with the national mortality data. Furthermore, a retrospective review of rehabilitation records from the Gross Motor Function Classification System (GMFCS) was conducted to assess the physical function maintenance rates of the residents. The GMFCS assessment scale was also used to evaluate residents’ physical function status and as an assessment reference for selecting assistive devices when individuals needed them in this nursing home. The study also involved interviews with staff members. In qualitative analysis, we designed simple interview questions to assess the staff’s understanding of epidemic-prevention measures and to evaluate their ability to demonstrate proper hand-washing and mask-wearing techniques. We used the results of this evaluation to develop an educational training program. The version of Microsoft Excel 2013 software analyzed the data, including calculating average and continuous percentages. The nursing home where the study was conducted has a capacity of 450 beds and comprises six residential areas. During the COVID-19 pandemic, 367 residents and 181 staff stayed in the nursing home. The guidelines of Taiwan’s CDC on infection control measures were implemented to prevent the spread of infection. Additionally, three health management measures were developed for nursing home residents. This was to maintain their quality of life, including health technology intervention, reablement program design for physical function rehabilitation, and video call equipment provided to maintain their social networks—usually, residents who can participate in rehabilitation activities in institutions function with a certain amount of physical activity.

Within the context of the GMFCS activity performance assessment, residents assigned to Level 1 could run, leap, and engage in everyday activities, such as walking and climbing stairs, without the need for assistive devices. Level 2 encompassed residents who could walk but occasionally required assistive devices for support. Level 3 was designated for residents who relied on assistance or used assistive devices while walking. It is worth emphasizing that individuals falling within the GMFCS assessment Levels 1 to 3 who retained their physical activity functions were included in this study to investigate their potential for reablement activities.

Residents unable to walk but capable of maintaining independent seated balance were categorized as Level 4, necessitating complete assistance. Conversely, residents who could not walk or maintain a seated balance and relied entirely on assistance were classified as Level 5 residents. It is crucial to underscore that individual at the GMFCS assessment Levels 4 and 5, characterized by a complete loss of physical function and irreversible deterioration of physical function, were excluded from this study (Table 1). A total of 105 residents participated in the reablement rehabilitation program to investigate whether reablement rehabilitation interventions during the pandemic could help to maintain residents’ physical activity levels and potentially delay the onset of functional decline, thereby extending their quality of life in terms of daily activities (Figure 1). 

### 2.2. Study Methods

All staff members were required to participate in an educational training program covering the guidelines for infection control, including implementing three health management interventions. This program included training staff on how to perform these interventions. The two phases of intervention programs were implemented, and data were collected from March 2021 through June 2022. This study was approved by the Research Ethics Committee (REC) of the Taichung Tzu Chi Hospital (No. REC111-47, 12 September 2022), and all participants were informed of the purpose of the study. 

Intervention One: The epidemic-prevention guidelines of the Taiwanese CDC are implemented. (1) Daily monitoring of respiratory symptoms and body temperature. If an individual shows symptoms of fever, they should be quarantined. (2) Follow a set of hand washing guidelines that include the following steps for proper hand hygiene: wash hands with soap for 40–60 s, emphasizing the palms, backs, between fingers, knuckles, thumbs, and fingertips, and then rinse hands. Alternatively, use hand sanitizer with alcohol and hand scrub for 20–30 s. It is important to wash hands at five key times: before meals, before and after any physical contact with others, after using the toilet, and after sneezing or coughing. (3) Wear a surgical mask to protect oneself from respiratory secretion splatter. (4) Maintain a social distance of 1.5 m. (5) Perform daily cleaning and disinfection of the environment [27]. (6) Restrict visitors during the COVID-19 pandemic.

Intervention Two: There are three measures in place for health management. (1) The health-technology-based measures include two components. The first component is a partnership with hospitals to provide remote telemedicine consultations for residents. The second component is a nursing monitoring notification system that takes the body temperature of all staff and residents twice daily using an infrared thermometer. (2) The staff assist residents in making good use of technology and social media networks daily, help residents stay connected with family and friends, and meet residents’ social needs. (3) Healthcare workers provide routine care to support residents’ physical function and facilitate reablement. This involves a range of exercises conducted in the seated or standing positions, along with practicing essential activities like getting out of bed, grasping objects, and achieving independent dressing. A rehabilitation therapist provides training instructions for various exercises, such as sit-to-stand, trunk forward reach, chest push-ups, and seated band row. The recommended routine is to perform each set of exercises 10 times, for 20–30 min per session, three sessions per week. Additionally, the therapist assesses residents’ progress using the GMFCS every three months and records the results.

## 3. Results

The majority (90%) of the residents in this study were over 65 years old, with 39.24% being over 85. The mean age of the residents was 79.4 years. There were 43.6% males and 56.4% females. Regarding the GMFCS level of assistance needs, 3% of residents required no assistance, 25.6% required some help, and 71.4% required complete aid. Additionally, all residents had multiple chronic diseases (Table 2).

Over 15 months, residents contracted COVID-19 without staff contracting the virus. Most cases occurred among individuals aged 71 to 90, accounting for 58.2% of confirmed cases. Among those infected, 69.1% were male and 30.9% were female. The overall infection rate varied between 2021 and 2022, but no outbreaks of nosocomial infections were reported. The mortality rate was low at 1.8%, with only one resident passing away due to COVID-19—a 90-year-old female with multiple chronic conditions. Confirmed cases received isolation care for an average of 17.7 days (Table 3).

The physical function of the residents was evaluated using the GMFCS, which showed that 9% and 22% of the residents experienced a decline in physical function in 2021 and 2022, respectively. Additionally, 36.55% and 11.4% demonstrated improvement, while 54.5% and 66.6% of residents maintained physical function in 2021 and 2022, respectively (Figure 2).

Regarding vaccination rates, the first and second vaccination rates for staff were both 100%. The first vaccination rate among residents was 96.5%, and the second was 94.6% (Figure 3).

## 4. Discussion

The findings of this study revealed that the COVID-19 mortality rate in our nursing home was 1.8%, which was lower than the national average mortality rate of 2.2% [7]. The study results indicate that frequent hand washing and surgical masks can prevent the spread of infectious diseases [28,29]. All staff and residents must follow the guidelines to prevent the spread of viruses and promote individual health. Staff members are a high-risk group due to frequent interactions with residents during the temporary suspension of visitations; it is vital to implement safety measures to ensure their well-being. To minimize the risk of COVID-19 transmission, staff members must undergo rigorous monitoring, including twice-daily temperature checks, regular symptom monitoring, and bi-weekly contact history reporting. Furthermore, they receive comprehensive infection-prevention training through e-learning programs to ensure that they are equipped to provide safe care for residents. The guidelines from Taiwan’s CDC emphasize the importance of social distancing, daily cleaning, visitor restrictions, isolation planning, and staff training to prevent the spread of disease. While limiting family visits can effectively prevent disease spread, it can also lead to social isolation, anxiety, and loneliness among residents. Moreover, the pandemic has severely restricted social activities and interactions, highlighting the need for epidemic-prevention measures and psychological support for residents. Maintaining an appropriate activity schedule is also essential in mitigating the negative effects of COVID-19 [30]. Providing video calls can be a viable solution to address the social needs of residents. This will facilitate regular communication between residents and their loved ones, fostering a sense of connectedness and reducing social isolation, especially when visits are restricted; video calls were utilized to reduce residents’ feelings of loneliness and social isolation [31]. Remote device intervention can promote residents’ positive attitudes and encourage emotional expression. Maintaining social interaction is essential for human well-being and can help prevent feelings of loneliness, isolation, and depression, particularly for older individuals living in long-term care facilities. Sustaining physical activity is critical to preserving the quality of life of long-term care residents during the pandemic, and an innovative approach called reablement training can effectively achieve this goal. Reablement training is incorporated into each individual’s daily routine and is designed to gradually improve their functional abilities as they engage in daily activities. This approach allows residents to build strength, increase mobility, and enhance their capacity to perform essential tasks. By tailoring the reablement training to each person’s unique needs and abilities, they can achieve their maximum potential and experience an improved quality of life [32]. Staff interviews revealed that some residents are experiencing pandemic-related anxiety, causing them to withdraw into their rooms and making it difficult to sustain social activities. Staff members are advised to be patient and encourage residents to maintain a daily routine and engage in activities beyond their rooms. The COVID-19 pandemic highlighted the need for innovative approaches to maintain long-term care residents’ physical and social activities. By incorporating reablement training into their rehabilitation activities, residents gradually became more willing to participate in their daily routine and engage in limited activities despite their limitations. An evaluation of residents’ physical function revealed that the physical function maintenance rate was 66.6%. These results demonstrate the effectiveness of reablement training in preserving residents’ physical function, enhancing their independence, and improving their overall well-being. Given the communal nature of their living arrangements, residents of long-term care facilities are particularly vulnerable to contracting SARS-CoV-2. As such, it is imperative that they receive early prioritization for vaccination to help to reduce the risk of transmission and keep this vulnerable population safe [33]. Based on the literature, vaccinations administered in long-term care facilities have demonstrated high success, with infection-avoidance rates varying between 53% and 92% [34,35,36]. This study encouraged staff and residents to get vaccinated, and the immunization rate was high, with 96.5% of residents and 100% of staff receiving the vaccine (Figure 3). Implementing epidemic-prevention measures necessitates the mobilization of resources and modifications to the environment. Nevertheless, visitors pose a challenge since they are an uncontrollable group and staff must allocate significant time and energy to educate them on pandemic protocols to reduce the risk of infection. Therefore, the shortage of resources and the increased demand for personnel during the epidemic are pressing concerns that require immediate attention.

## 5. Limitations

This study has certain limitations and challenges. Implementing epidemic-prevention measures necessitates the mobilization of resources and modifications to the environment. Entry and exit routes must be re-planned to implement the Taiwanese CDC’s epidemic-prevention guidelines and reduce intensive interaction between people. Additionally, separate space partitions are necessary to provide isolation rooms for confirmed cases. In addition, the planning of reablement rehabilitation activities must be redesigned according to the remodeling of the space. The biggest challenge is that all space transformations must be completed in a very short time. Nevertheless, visitors are another challenge since they are an uncontrollable group and staff must allocate significant time and energy to educate them on pandemic protocols to reduce the risk of infection. The methods and habits of washing hands correctly and wearing masks need to be established within a short period of time, and some residents may not be willing to cooperate. The high demand for manpower poses a challenge in implementing these three health management measures. Therefore, the shortage of resources and the increased demand for personnel during the epidemic are pressing concerns that require immediate attention. In future studies, it would be beneficial to explore the effectiveness of different training and education programs for staff and visitors in nursing homes, such as in-person training sessions versus online training modules. Regarding the workforce shortage, we recommend nursing homes explore innovative solutions to address these challenges for future study. One potential solution is to import newer technology to fill the gaps in staffing and resources. 

## 6. Conclusions

Managing COVID-19 infections in long-term care facilities is challenging. However, effective guidelines for epidemic-prevention measures can help to prevent outbreaks and control the spread of the disease. In this study, we implemented the “Guidelines on Infection Control Measures for Long-Term Care and Social Welfare Institutions in Response to Severe Special Infectious Pneumonia” provided by Taiwan’s CDC. This effective infection control strategy can be used as a reference for epidemic prevention. While restrictive measures could increase anxiety and loneliness among residents, utilizing social interactive media and regular activities can help to reduce the incidence of depression in older people. In our study, during the pandemic, social software was used to provide residents with external contact and social interaction to reduce their depression. Modern technology continues to innovate and develop new software. In the future, we will study whether social media applications can effectively reduce the occurrence of depression among institutional residents. In addition, we applied reablement rehabilitation to effectively maintain the residents’ physical activity and overall well-being. This illustrates that integrating simple daily rehabilitation activity designs into residents’ lives can effectively maintain or delay the deterioration of their physical function. Based on the above research results, it is important to continue exploring innovative solutions to address the challenges faced by long-term care facilities during the pandemic.

Long-term care facilities should maintain elderly residents’ hygiene and lifestyle while adhering to government-mandated pandemic prevention rules to improve institutional caregivers’ ability to prevent epidemics. Tight control of external sources of infection should be implemented by restricting institutional visitors, referrals of suspected cases, and interactions with the local community. It is necessary to remodel settings and operational procedures to ensure the safety of residents and workers during a pandemic. The consensus among the staff is the priority. Therefore, it is recommended that all staff members receive education and training. Our experience indicates that every medical, psychological, and physical care component should be considered when implementing infection-prevention measures. Institutions can use this experience to prepare for future pandemic emergencies and prevent them from occurring. Reablement, which has shown promising results in this study, is commonly utilized during in-home rehabilitation. Long-term care facilities may benefit from developing a comprehensive reablement plan and implementing appropriate evaluation methods as part of their activity rehabilitation design schemes for residents. This can help to promote functional independence, enhance overall well-being, and improve the quality of life for elderly individuals in long-term care settings.

This study highlights the need for effective epidemic-prevention guidelines that provide clear directions for people to follow during an outbreak. Proper hand washing and the use of surgical masks have proven effective in controlling infections. Even outside of epidemic situations, long-term care facilities should promote good hand-hygiene practices among staff and residents to reduce infection rates. In health management, incorporating rehabilitation plans into residents’ daily activities is essential for encouraging active participation. Additionally, the ever-evolving field of technology plays a crucial role in healthcare and long-term care. Although technology cannot replace human interactions, it can serve as a valuable tool for interpersonal communication. Healthcare and long-term care facilities can use diverse technological applications in the care sector to enhance the quality of care.

## Figures and Tables

**Figure 1 healthcare-11-02535-f001:**
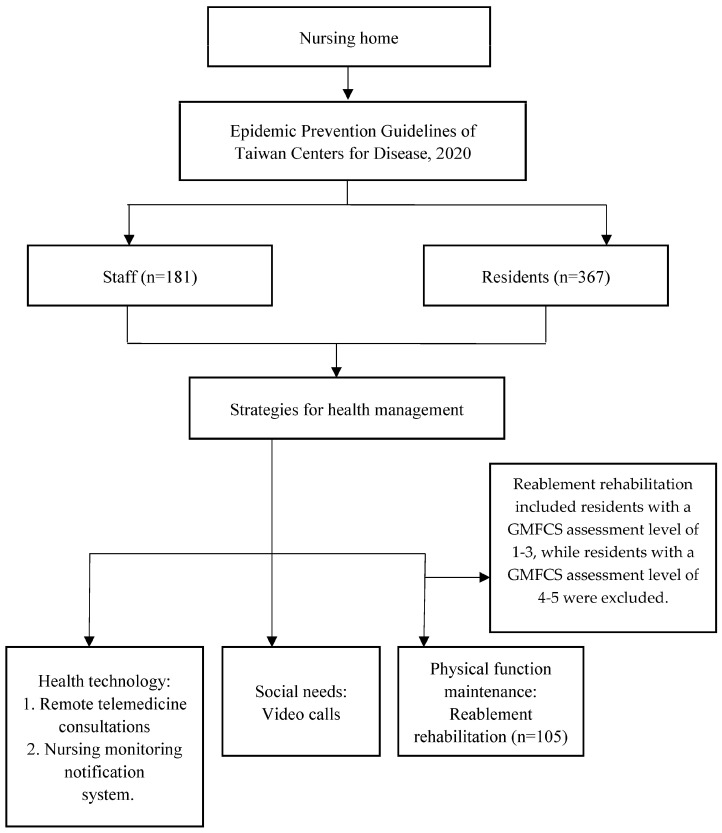
Flow diagram of the study.

**Figure 2 healthcare-11-02535-f002:**
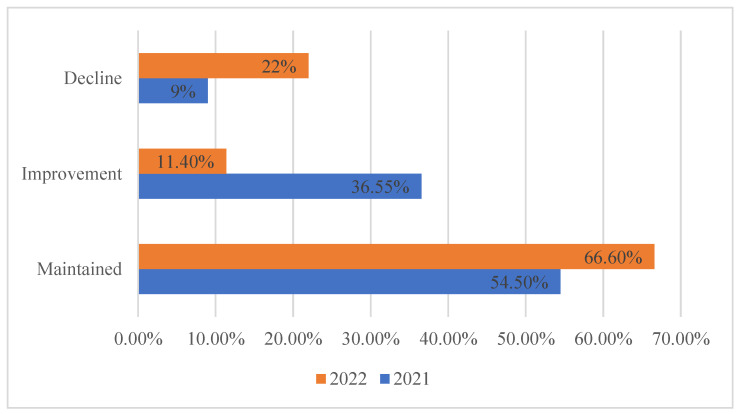
Residents’ physical function maintenance rate.

**Figure 3 healthcare-11-02535-f003:**
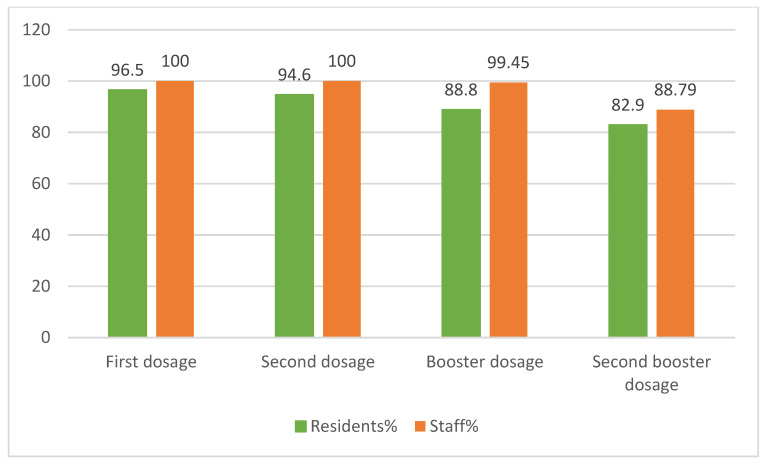
The nursing home vaccination rate.

**Table 1 healthcare-11-02535-t001:** GMFCS activity levels.

GMFCS Levels	Activity Performance Evaluation
Level 1	Able to run, leap, and perform everyday activities like walking and climbing stairs without a handhold. Further, they require no assistance.
Level 2	Can walk on a flat surface but sometimes needs assistive devices.
Level 3	Requires assistance or assistive devices when walking.
Level 4	Cannot stride and walk, but able to maintain balance in a sitting position independently. Complete assistance needed.
Level 5	Unable to walk or maintain balance in a sitting position. Complete assistance needed.

GMFCS: Gross Motor Function Classification System [25,26].

**Table 2 healthcare-11-02535-t002:** Demographic characteristics of residents (N = 367).

Demographic Characteristics		Profile	n	%
Age		<54	17	4.63
		55–64	21	5.72
		65–74	66	17.98
		75–84	119	32.43
		>85	144	39.24
Mean age		79.4		
Gender		Male	160	43.6
		Female	207	56.4
Chronic disease		Yes	367	100
		No	0	0
GMFCS Level	Level 1	Requires no assistance	11	3.0
	Level 2–3	Some assistance needed	94	25.6
	Levels 4–5	Complete assistance needed	262	71.4

**Table 3 healthcare-11-02535-t003:** Nursing home COVID-19 confirmed cases (N = 55).

Demographic Characteristics	Profile	n	%	Days
Age	35–50	4	7.3	
	51–70	12	21.8	
	71–90	32	58.2	
	>90	7	12.7	
Gender	Male	38	69.1	
	Female	17	30.9	
Vaccinated	Yes	51	92.7	
	no	4	7.3	
COVID-19	Diagnosed Severe	55 2	10 3.6	
	Death	1	1.8	
Outbreak control	Average quarantine days			17.7

## Data Availability

The data presented in this study are available upon request from the corresponding author.

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
