# Peer review of "Epidemic-Prevention Measures and Health Management in a Nursing Home during the Coronavirus Disease 2019 Pandemic"

_healthcare, 2023, doi:10.3390/healthcare11182535_

Round 1
Author Response
Dear Reviewer,
Thank you for your valuable time and effort in reviewing our manuscript. Your feedback has been very beneficial to us. We have addressed your helpful and insightful points with responses in blue with a highlight. please see the PDF file.

Reviewer 2 Report
This study presents the impact of epidemic prevention and containment policies on the health and well-being of patients and evaluates the effectiveness of implementing intervention measures to maintain their quality of life.
In this paper, the authors investigate the measures to prevent the COVID-19 epidemic that can help prevent outbreaks and control the spread of the disease.
I would raise some problems that I suggest the authors solve for the clarity of the paper and a more significant impact in the domain:
· The authors mention that “the mixed methods” were used in the research. What kind of methods, these methods should be clearly pointed out.
· The results of the study indicate that "frequent hand washing and surgical masks can prevent the spread of infectious diseases", Other conclusions and findings of the study would be of great interest, to complement what is already known and demonstrated.
· In the Discussion section, the results could be presented in a graphical/tabular form for better understanding by the readers, highlighting the key elements.
· In Conclusion, it is mentioned that innovative solutions are needed to address the challenges faced by long-term care facilities during the pandemic. In this sense, I recommend that the authors clearly highlight the solutions proposed by their research and be mentioned both in the abstract and in the conclusions.
· Figure 1. Flow diagram of the study - review the lines between boxes.
Author Response
Dear Reviewer,
Thank you for your valuable time and effort in reviewing our manuscript. Your feedback has been very beneficial to us. We have addressed your helpful and insightful points with responses in blue with a highlight. Please review the PDF document. Thank you so much.

Reviewer 3 Report
I would like to thanks the authors for their work.
This is an interesting paper, which aims to investigate the impact of epidemic prevention and isolation policies on residents’ health and well-being and assess the effectiveness of implementing intervention measures to maintain their quality of life in Taiwan.
The abstract is well-written and informative.
Some minor aspects can improve the readability of the introduction:
- Page 2, lines 50-51, "Taiwan’s overall mortality rate was 0.17%, which was relatively low compared to many other countries": I suggest to add the reference for an easier comprehension
- Page 2, lines 55-56, "Nursing homes are recognized as having a high risk of COVID-19 infection due to their residents’ advanced age and often multiple chronic conditions": I suggest to add the reference for an easier comprehension
- Page 2, lines 84-85, "It is essential to prioritize optimizing opportunities for health participation and security to improve the quality of life for aging adults": I suggest to add the reference for an easier comprehension
- Nevertheless, the "transition" between the COVID-19 and the opportunities for health participation (page 2, lines 83-84) is very stark; I would better clarify the link between these two concepts
The methods are clear, but I would suggest to revise the graphic rendering of Figure 1, the lines appear disordered.
The results are clear and consistent with the declared methodology.
Discussion:
- Page 7, line 214: is it possible to calculate the p-value of the difference between the nursing home mortality rate (1.8%) and the national mortality rate? Even if the paper is only descriptive, this information can lead to a stronger interpretation of this difference
- Page 7, lines 214-215: "...and the estimated mortality rate (8.4%) in the United States": as the US setting is completely different, the comparison between the rates can result inappropriate; I suggest to remove this part or, alternatively, to argue this comparison in the text
The conclusions are clear and not speculative.
The bibliography is appropriate and up-to-date
Author Response

(The authors gave the same response as above.)

Round 2
Reviewer 2 Report
Dear authors,
I appreciate the effort and the modified manuscript according to my suggestions.
I wish you continued success.